

**Can the Nucleation Phase be Generated on a Sub-fault**
**Linked to the Main Fault of an Earthquake?**
Jeen-Hwa Wang
Institute of Earth Sciences, Academia Sinica , P.O. Box 1-55, Nangang, Taipei,
TAIWAN email: jhwang@earth.sinica.edu.tw
(submitted toNonlinear Processes in Geophysics on November 3, 2018)
**Abstract**. We study the effects of seismic coupling, friction, viscous, and inertia on
earthquake nucleation based on a two-body spring-slider model in the presence of
thermal-pressurized slip-dependent friction and viscosity. The stiffness ratio of the
system to represent seismic coupling is the ratio of coil spring $K$ between two sliders
and the leaf spring $L$ between a slider and the background plate and denoted by $s=K/L$.
The $s$ is not a significant factor in generating the nucleation phase. The masses of the
two sliders are $m_1$ and $m_2$, respectively. The frictional and viscous effects are
specified by the static friction force, $f_o$, the characteristic displacement, $U_c$, and
viscosity coefficient, $\eta$, respectively. Numerical simulations show that friction and
viscosity can both lengthen the natural period of the system and viscosity increases
the duration time of motion of the slider. Higher viscosity causes lower particle
velocities than lower viscosity. The ratios $\gamma=\eta_2/\eta_1$, $\phi=f_{o2}/f_{o1}$, $\psi=U_{c2}/U_{c1}$, and
$\mu=m_2/m_1$ are four important factors in influencing the generation of a nucleation
phase. When $s>0.17$, $\gamma>1$, $1.15>\phi>1$, $\psi<1$, and $\mu<30$, simulation results exhibit the
generation of nucleation phase on slider 1 and the formation of $P$ wave on slider 2.
The results are consistent with the observations and suggest the possibility of

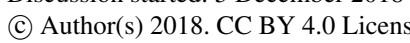
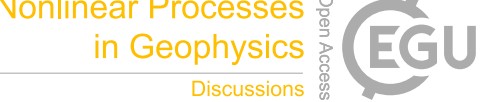


generation of nucleation phase on a sub-fault.
**Keywords:** nucleation phase, two-body spring-slider model, stiffness ratio, thermal-
pressurized slip-dependent friction, viscosity



## 1 **Introduction**

The presence of nucleation phase before the *P* waves (see Fig. 1) was suggested by early theoretical studies (e.g., Andrews, 1976; Brune, 1979; Dieterich, 1986, 1992; Das and Scholz, 1981) and laboratory experiments (Dieterich, 1979; Ohnaka et al., 1987). Some studies (Scholz et al., 1972; Dieterich, 1981; Ohnaka and Yamashita, 1989; Ohnaka, 1992; Ohnaka and Kuwahara, 1990; Kato et al., 1994; Roy and Marone, 1996; Lu et al., 2010; Latour et al., 2013; Kaneko et al., 2016) also indicated that the nucleation process behaves like a transition from quasi-static slip (without the inertial effect) to (unstable) dynamic motion (with the inertial effect) when the slip speeds become high enough to make the inertial effect dominate frictional resistance under some conditions. The study of this phase is a basic problem of earthquake physics and also important for early warming, prediction, and hazard assessment of earthquakes.

Umeda (1990) first recognized the nucleation phase in velocity seismograms. Since then, numerous seismologists also observed the nucleation phases (e.g., Iio, 1992, 1995; Ellsworth and Beroza, 1995; Beroza and Ellsworth, 1996; Mori and Kanamori, 1996; Ruiz et al., 2017). There is a debate concerning the correlation between the duration time, $T_D$, of nucleation phase and the magnitude, $M$, of the earthquake occurring immediately after the nucleation phase. Ellsworth and Beroza (1995) and Beroza and Ellsworth (1996) assumed a positive correlation of $T_D$ to $M$. Whereas, Mori and Kanamori (1996) observed independence of the *P* waves on the shape of nucleation phase in a large magnitude range. Ellsworth and Beroza (1998) confirmed the observation by Mori and Kanamori (1996).

Friction and viscosity are two major factors in controlling the complicated earthquake rupture processes including nucleation (see Wang, 2016; and cited



references therein). Analytic solutions and numerical simulations for exploring the
nucleation phase have made based on the infinite dislocation models, crack models,
and spring-slider models by using different friction laws (see Beeler, 2004; Tal et al.,
2018; Wang, 2016, 2017a; and cited references therein). Iio (1992, 1995) stressed that
the nucleation phase cannot be interpreted by any theoretical source model with a
constant kinematic friction and a constant rupture velocity. Mori and Kanamori (1996)
claimed that any model having a similar initial rupture can describe the nucleation
phases of earthquakes of all sizes, and thus it is difficult to estimate the magnitude of
an earthquake just from its nucleation phase. They also stressed that curvature seen in
the nucleation phases is caused by anelastic attenuation.

Some theoretical studies based on the Burridge-Knopoff spring-slider model

(Burridge and Knopoff, 1967), from which the two-body model used in this study is
simplified, are briefly described here. Brantut et al. (2011) concluded that
metamorphic dehydration influences the nucleation of unstable slip and could be an
origin for slow-slip events in subduction zones. Ueda et al. (2014, 2015) and
Kawamura et al. (2018) pointed out that the nucleation process includes the quasi-
static initial phase, the unstable acceleration phase, and the high-speed rupture phase
(i.e., a mainshock) and recognized two kinds of nucleation lengths, i.e., $L_{sc}$ and $L_c$
which are affected by model parameters, yet not by the earthquake size. The $L_{sc}$
related to the initial phase exists only for a weak frictional instability regime; while
the $L_c$ associated with the acceleration phase exist for both weak and strong instability
regimes. They also found that in the initial phase up to $L_{sc}$, the sliding velocity is of
order the plate speed, while at a certain stage of the acceleration phase it becomes
higher and thus can be observed.

Although the frictional effect on earthquake nucleation has been long and widely



studied as mentioned above, the studies of viscous effect on earthquake ruptures are
rare. The viscous effect mentioned in Rice et al. (2001) was actually an implicit factor
which is included within the direct effect of rate- and state-dependent friction law.
Wang (2017a) took viscosity into account for studying the nucleation phase by
assuming a temporal change of high viscosity to low viscosity during an earthquake
rupture based on a one-body spring slider model with thermal-pressurized slip-
weakening friction. His results in a temporal variation from nucleation phases to $P$
wave and the amplitude of $P$ wave, which is associated with the earthquake
magnitude, does not depend on the duration time of the former.

As mentioned above, the nucleation process behaves like a transition from quasi-

static slip (without the inertial effect) to (unstable) dynamic motion (with the inertial
effect) when the slip speeds become high enough to make the inertial effect dominate
frictional resistance under some conditions. This assumes that the inertial effect must
be taken into account.

In most of studies, both the nucleation phase and the $P$ wave are assumed to

occur on the same fault. There is an interesting question: Can the nucleation phase
happen on a sub-fault which links to the main fault of an earthquake? In order to
answer this question, in this work I will explore the frictional, viscous, and inertial
effects on the generation of nucleation phase on a fault and then the transition from it
to the $P$ wave on the other based on a two-body spring-slider model, which is used to
approach an earthquake fault (see Galvanetto, 2002; Turcotte, 1992), by considering
the two sliders to be two segments of an earthquake fault,. The friction force caused
by thermal pressurization is slip-weakening and the viscosity is represented by an
explicit parameter. In addition, it is significant to consider the inertial effect on the
earthquake nucleation because of the existence of transition from quasi-static motion

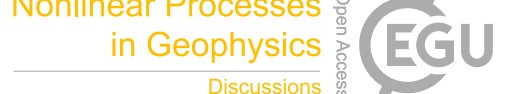

to dynamic ruptures from observations and laboratory experiments. The study on
inertial effect on nucleation phase is rare, even though this effect is implicitly
included in the thermal-pressurized friction used by Brantut et al. (2011). Here, the
inertial effect will be taken into account.

**2 Two-body Spring-slider Model**
The two-body spring-slider model (Fig. 2) consists of two sliders of mass $m_i$ ($i$=1, 2)
and three springs. The detailed description of the model can be seen in Wang (2017b)
and only briefly explained here. The equation of motion of the system is

$$m_1 d^2 u_1/dt^2 = K(u_2-u_1)-L_1(u_1-v_p t)-F_1(u_1)-\Phi(v_1) \qquad (1a)$$

$$m_2 d^2 u_2/dt^2 = K(u_1-u_2)-L_2(u_2-v_p t)-F_2(u_2)-\Phi(v_2). \qquad (1b)$$

The $u_i$ ($i$=1, 2) is the displacement of the slider measured from its initial equilibrium
position along the x-axis. The $K$ is the strength of the coil spring between two sliders
and the $L_i$ ($i$=1, 2) is the strength of the leaf spring to yield the driving force on the
$i$-th slider from a moving plate with a constant speed $v_P$. Considering the two sliders
to be two segments of a single earthquake fault, the coupling between the moving
plate and a slider could be equal for the two sliders, thus giving $L_1=L_2=L$. $F_i(u_i)$ ($i$=1,
2) is the frictional force on the $i$-th slider. Wang (2013) took $F(u)=F_o exp(-u/u_c)$,
where $F_o$ and $u_c$ are, respectively, the static friction force and characteristic slip
displacement, to study earthquake dynamics. This friction force is slip-weakening and
caused by the adiabatic-undrained-deformation (AUD)-type thermal pressurization
(Rice, 2006). An example of the variations of $F(u)$ versus $u$ for $F_o$=1 N and $u_c$=0.1,



0.3, 0.5, 0.7, and 0.9 m is displayed in Fig. 3. *F(u)* decreases with increasing u, and
the decreases rate is higher for smaller $u_c$ than for larger $u_c$. This indicates that the
force drop decreases with increasing $u_c$ for the same final displacement. The *Φ(v$_i$),*
where $v_i = du_i/dt$ is the particle velocity, is a velocity-dependent viscous force.
According to Stokes' law, Wang (2016) suggested the viscous force to be *Φ=Cv,*
where $C = 6\pi R\upsilon$ (with a unit of $N(m/s)^{-1}$) is the damping coefficient of a sphere of
radius *R* in a fluid of viscosity $\upsilon$ (Kittel et al. 1968). The two sliders rest in an
equilibrium state at time *t=0*. Note that this model addresses only the strike-slip
component and, thus, cannot completely represent earthquake ruptures, which also
consist of transpressive components. Nevertheless, simulation results of this model
can still provide significant information on earthquake ruptures.

Substituting the friction and viscous laws into Equation (1) leads to


$m_1 d^2 u_1/dt^2 = K(u_2-u_1) - L(u_1-v_P t) - F_{o1} exp(-u_1/u_{c1}) - C_1 du_1/dt$          (2a)


$m_2 d^2 u_2/dt^2 = K(u_1-u_2) - L(u_2-v_P t) - F_{o2} exp(-u_2/u_{c2}) - C_2 du_2/dt$          (2b)


To deal with the problem easily, it is usual to normalize Equation (2) based on the
normalization parameters. Wang (1995) defined the stiffness ratio, *s*, to be the ratio of
*K* to *L*, i.e., *s=K/L*. Wang (2017b) defined the normalization parameters for Equation
(2). However, in his study he took $m_1=m_2$, and thus he did not consider the cases with
different values of $m_1$ and $m_2$. While, in this study $m_2$ could be larger than $m_1$ for
showing the inertial effect. Hence, the parameters normalizing Equation (2) are:
$m_1=m$, $m_2=\mu m$, $F_{o1}=F_o$, $F_{o2}=\phi F_o$, $D_o=F_o/L$, $\omega_{o1}=\omega_o=(L/m)^{1/2}$, $\omega_{o2}=\mu^{-1/2}\omega_o$, $\tau=\omega_o t$,
$u_{c1}=u_c$, $u_{c2}=\psi u_c$, $U_{c1}=u_c/D_o$, $U_{c2}=\psi u_c/D_o$, $f_{o1}=f_o=F_o/D_o$, $f_{o2}=\phi f_o$, $\eta_1=C_1\omega_o/L$,





$\eta_2=C_2\mu^{-1/2}\omega_o/L$, $\gamma=\eta_2/\eta_1$, and $V_P=v_P/D_o\omega_o$. Defining $U_i=u_i/D_o$ and $V_i=dU_i/d\tau$ leads to
$du_i/dt=[F_o/(mL)^{1/2}]dU_i/d\tau$ and $d^2u_i/dt^2=(F_o/m)d^2U_i/d\tau^2$. Inserting these normalization
parameters with $f_o=1$ into Equation (2) results in:

$d^2U_1/d\tau^2=s(U_2-U_1)-(U_1-V_P\tau)-exp(-U_1/U_{c1})-\eta_1dU_1/d\tau$        (3a)


$d^2U_2/d\tau^2=[s(U_1-U_2)-(U_2-V_P\tau)-\phi exp(-U_2/U_{c2})-\eta_2dU_2/d\tau]/\mu.$        (3b)


Let $y_1=U_1$, $y_2=U_2$, $y_3=dU_1/d\tau$, and $y_4=dU_2/d\tau$. Equation (3) can be re-written

as four first-order differential equations:

$dy_1/d\tau=y_3$                                (4a)


$dy_2/d\tau=y_4$                                (4b)


$dy_3/d\tau=-(s+1)y_1+sy_2-exp(-y_1/U_{c1})-\eta_1y_3+V_P\tau$        (4c)


$dy_4/d\tau=[sy_1-(s+1)y_2-\phi exp(-y_2/\psi U_{c1})-\gamma\eta_1y_4+V_P\tau]/\mu.$        (4d)


Since it is difficult to analytically solve Equation (4), only numerical simulations
using the fourth-order Runge-Kutta method (see Press et al., 1986) is performed in
this study. Note that the sliders are restricted to move only along the positive direction,
that is, $V_i\geq0$ and $U_i\geq0$ (i=1, 2).

**3 Numerical Simulations**

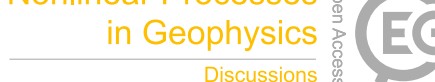

Before performing numerical simulations, it is necessary to consider the acceptable
values of model parameters. Strong coupling can make the two sliders move almost
simultaneously. Hence, in order to allow independent motion for each slider, the value
of $s$ should be small. Numerical tests (Wang, 2017b) show weak coupling as $s<5$ and
strong coupling as $s \geq 5$ for a two-body spring-slider system. Hence, $s<5$ is considered
in this study. In general, $v_P$ is $\sim 10^{-9}$ m/s and thus $V_P$ is $\sim 10^{-9}$ when $D_o \omega_o$ is an order of
magnitude of 1 m/sec. Simulation results could be influenced by using various time
steps, $\delta \tau$. Practical tests suggest that simulation results show numerical stability when
$\delta \tau < 0.05$. The time step is taken to be $\delta \tau = 0.02$ hereafter. When $V_P \tau = exp(-y_1/U_{c1})$ on
slider 1 from Equation (4c), the force exerted from the moving plate is just equal to $f_{oi}$.
Although in principle slider 1 can start to move under this condition, in practice the
computation cannot go ahead because all values are zero. An initial force, $\delta f$, is
necessary to kick off slider 1. Note that the value of $\delta f$ can affect the computational
results (Carlson et al., 1991). A very small value of $\delta f$ cannot enforce slider 1 to move;
while a large one will dominate the whole computation process. Numerical tests show
that $\delta f = 10^{-3}$ is appropriate for numerical simulations.

Numerical simulations are made under various values of model parameters for

showing the effects caused by seismic coupling, friction, viscosity, and inertial effect.
Simulation results are displayed in Figures 4−10 which include the time variations in
$V/V_{max}$ (in the left-hand-side panels) and $U/U_{max}$ (in the right-hand-side panels).

The results for the effect due to seismic coupling are displayed in Fig. 4 where

the values of $s$ are: (a) for $s=0.06$, (b) for $s=0.12$, (c) for $s=0.30$, and (d) for $s=0.48$
when $f_{o1}=1.0$ and $f_{o2}=1.0$ (with $\phi=1$), $U_{c1}=0.5$ and $U_{c2}=0.5$ (with $\psi=1$), and $\eta_1=0$ and
$\eta_2=0$ (with $\gamma=1$). First, it is necessary to examine the lower-bound value of $s$ for
yielding strong enough coupling between the two sliders. Numerical tests exhibit that




slider 2 cannot move for $s<0.06$ when other model parameters are equal on the two
sliders. Hence, $s=0.06$ is almost the lower bound of seismic coupling for most of
simulations. On the other hand, numerical tests suggest that when $s>0.48$, the solid
and dashed lines are coincided. This means that large $s$ having strong seismic
coupling leads to almost simultaneous motions of the two sliders. Hence, the value of
$s$ is taken to be 0.48 in Figs. 5−7, and 9 to explore which factor can separate the
motions of the two sliders.

Figures 5−8 display the results due to different values of viscosity on the two

sliders when other parameters are fixed: (a) for $\gamma=0.00$ (i.e., $\eta_2=0$), (b) for $\gamma=0.01$ (i.e.,
$\eta_2=0.1$), (c) for $\gamma=0.05$ (i.e., $\eta_2=0.5$), and (d) for $\gamma=0.10$ (i.e., $\eta_2=1$) when $\eta_1=10$. In
Fig. 5 the values of other model parameters are $\mu=1$, $\eta_1=10$, $s=0.48$, $f_{o1}=1.0$ and
$f_{o2}=1.0$ (with $\phi=1$), and $U_{c1}=0.5$ and $U_{c2}=0.5$ (with $\psi=1$). The figure displays the
presence of the $P$ wave on slider 2. Numerical tests reveal that the $P$ wave on slider 2
cannot be generated especially for $\gamma \geq 0.05$ when $\eta_1>70$, and the solutions are just like
Figure 4 when $\eta_1<5$. Hence, $\eta_1$ is taken to be 10 in Figs. 6−10. The simulation results
to exhibit the effect due to different static friction strengths on the two sliders, are
displayed in Fig. 6, where the values of other model parameters are $\mu=1$, $\eta_1=10$,
$s=0.48$, $f_{o1}=1.0$ and $f_{o2}=1.1$ (with $\phi=1.1$), and $U_{c1}=0.5$ and $U_{c2}=0.5$ (with $\psi=1$). The
figure exhibits the presence of a nucleation phase on slider 1. Numerical tests exhibit
that when $\phi>1.15$, the $P$ wave on slider 2 cannot be generated. Hence, $\phi$ is taken to be
1.1 in Figs. 7−10. The simulation results to exhibit the effect due to different
characteristic displacements on the two sliders are displayed in Fig. 7, where the
values of other model parameters are $\mu=1$, $\eta_1=10$ $s=0.48$, $f_{o1}=1.0$ and $f_{o2}=1.1$ (with
$\phi=1.1$), and $U_{c1}=0.5$ and $U_{c2}=0.1$ (with $\psi=0.2$). The figure shows the presence of a
nucleation phase on slider 1. Numerical tests exhibit that when $U_{c1}>0.5$, the $P$ wave

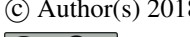

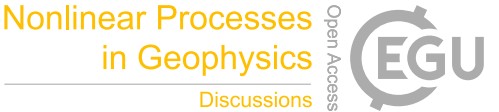

on slider 2 cannot be generated. Hence, $U_{c1}$ is taken to be 0.5 in Figs. 8−10. In order
to consider weaker seismic coupling on the simulated waveforms, smaller s is taken
into account. Numerical tests exhibit that when $s<0.17$, the $P$ wave on slider 2 cannot
be generated. Hence, $s$ is also taken to be 0.17 in Fig. 8 where the values of other
model parameters are $\mu=1$, $\eta_1=10$, $s=0.17$, $f_{o1}=1.0$ and $f_{o2}=1.1$ (with $\phi=1.1$), and
$U_{c1}=0.5$ and $U_{c2}=0.1$ (with $\psi=0.2$).

Figures 9 and 10 display the results for the inertial effect due to different masses

of the two sliders: (a) for $\mu=1$, (b) for $\mu=5$, (c) for $\mu=10$, and (d) for $\mu=30$ when $\eta_1=10$,
$f_{o1}=1.0$ and $f_{o2}=1.1$ (with $\phi=1.1$), $U_{c1}=0.5$ and $U_{c2}=0.1$ (with $\psi=0.2$), and $\eta_1=10$ and
$\eta_2=0$ (with $\gamma=0$). The main difference between the two figures is the use of different
values of seismic coupling: $s=0.48$ in Fig. 9 and $s=0.17$ in Fig. 10.

In the panels of Figs. 4−10, the simulation results for slider 1 and slider 2 are

represented, respectively, by a solid line and a dotted line. Numerical results show that
the values of $V_{max}$ and $U_{max}$ are: 0.456 and 1.355, respectively, in Fig. 4; 0.142 and
0.798, respectively, in Fig. 5; 0.226 and 0.766, respectively, in Fig. 6; 0.781 and 1.403,
respectively, in Fig. 7; 0.903 and 1.778, respectively, in Fig. 8; 0.781 and 1.505,
respectively, in Fig. 9; and 0.903 and 1.790, respectively, in Fig. 10.

**4 Discussion**
4.1 Seismic Coupling Effect
Figure 4 shows the simulation results when $s=0.06$, 0.12, 0.30, and 0.48 (upside
down). In the left-hand-side panels for $V/V_{max}$, the dashed line separates from the
solid line for small $s$, while the two lines are almost coincided for large $s$. This reflects
the fact that seismic coupling between the two sliders increases with $s$. Meanwhile,
the peak amplitude is larger at slider 2 than at slider 1. This is reasonable due to the

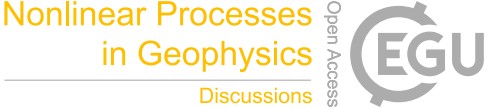

directivity effect because the system moves from slider 1 to slider 2. However, this
figure does not exhibit the existence of long-period nucleation phase. Hence, seismic
coupling is not a significant factor in the generation of nucleation phase. From the
right-hand-side panels for $U/U_{max}$, we can also obtain the same conclusion as
mentioned above. In addition, the final displacements on the two sliders are almost
equal.

4.2 Viscous Effect
Simulation results based on a one-body spring-slider model by Wang (2017a) show
that a change of viscosity from a lager value to a small one in two time stages during
slippage yields the nucleation phase and the $P$ wave, respectively, in the first and
second stages. Hence, in Fig. 5 the value of $\eta_1$ is set to be 10 and $\eta_2$ varies from 0 to 1
or $\gamma$ varies from 0.0 to 0.1 for $s$=0.48 when the values of other parameters are the
same as those in Fig. 4. The left-hand-side panels of Fig. 5 exhibit the presence of a
short-time nucleation phase plus a smaller event on slider 1 and a larger event with a
$P$ wave on slider 2. Hence, there are two sub-events during the whole rupture process.
The peak velocity of slider 2 decreases with increasing $\gamma$, yet not for slider 1. The
peak velocity appears earlier on slider 2 than on slider 1. The occurrence time of the
peak velocity of slider 2 slightly increases with $\gamma$. In addition, there are few events
with low peak velocities after the main one on slider 2, and the number of small
events decreases with increasing $\gamma$.

The predominant period and the peak velocity of slider 1 are, respectively, longer

and smaller than those of slider 2. Of course, the differences decrease with increasing
$\gamma$ or $\eta_2$. Compared with Fig. 4, the predominant periods for the two sliders in Fig. 5
become longer due to the viscous effect. From Equation (1), the (dimensionless)

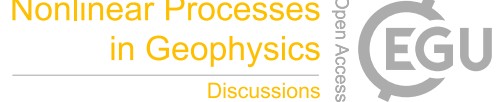

natural period is $T_{o1}=T_o=2\pi(m/L)^{1/2}$ for slider 1 and $T_{o2}=2\pi(\mu m/L)^{1/2}=\mu^{1/2}T_o$ for
slider 2 when the two sliders are not linked together and friction and viscosity are
both absent. When the two sliders are linked together, the natural period of each slider
must be slightly different from $T_{o1}$ or $T_{o2}$. When viscosity is included, the natural
period    is    $T_1=T_{o1}/(1-C_1^2/4mL)^{1/2}=T_o/(1-\eta_1^2/4)^{1/2}$    for    slider    1    and
$T_2=T_{o2}/(1-C_2^2/4\mu mL)^{1/2}=\mu^{1/2}T_o/(1-\eta_2^2/4)^{1/2}$ for slider 2. Obviously, viscosity
increases the natural period of oscillations of each slider and also depresses the peak
velocity. The ratio of $T_2$ to $T_1$ is:

$$T_2/T_1=[\mu(4-\eta_1^2)/(4-\eta_2^2)]^{1/2}=[\mu(4-\eta_1^2)/(4-\gamma\eta_1^2)]^{1/2}.\qquad(5)$$


Equation (5) shows that when $\mu>1$ and $\gamma>1$, we have $T_2>T_1$. When $\eta_2$ approaches 2,
$T_2$ becomes infinity. Hence, $\eta_2=2$ is an upper bound of generating a normal $P$ wave.
The left-hand-side panels of Fig. 5 exhibits an increase in $T_2$ with $\eta_2$ or $\gamma$.

In the right-hand-side panels of Fig. 5, the displacement of slider 2 (displayed by

a dashed line) first increases more rapidly than that of slider 1 (shown by a solid line)
and finally two lines merge together, thus exhibiting the same final displacement on
the two sliders.

Although we can see the existence of long-period waveform on slider 1 in Fig. 5,

its peak velocity comes after that of a short-period $P$ wave on slider 2. This does not
exhibit transition from quasi-static motions to dynamic ruptures as shown from
observations, and thus the whole waveform on slider 1 cannot be classified to be the
nucleation phase. Hence, it is assumed that different values of viscosity coefficients
on the two sliders are not the unique factor to yield the nucleation phase for the
two-body model, and thus the differences in other model parameters between the two



sliders must be taken into account.

4.3 Frictional Effect
The fictional effect includes two components: the static friction forces or the frictional
strength (denoted by $f_{o1}$ and $f_{o2}$ on slider 1 and slider 2, respectively) and the
characteristic displacements of friction law (represented by $U_{c1}$ and $U_{c2}$ on slider 1
and slider 2, respectively). First, we consider different values of $s$, $f_{o1}$, and $f_{o2}$.
Simulation results are displayed in Fig. 6 where static friction forces are $f_{o1}$=1.0 and
$f_{o2}$=1.1 (with $\phi$=1.1) when other values of model parameters are the same as those in
Fig. 5, i.e., $s$=0.48, $U_{c1}$=0.5 and $U_{c2}$=0.5 (with $\psi$=1), and (a) for $\gamma$=0.00 or $\eta_2$=0, (b)
for $\gamma$=0.01 or $\eta_2$=0.1, (c) for $\gamma$=0.05 or $\eta_2$=0.5, and (d) for $\gamma$=0.10 or $\eta_2$=1 when $\eta_1$=10.
The left-hand-side panels show the presence of a very long-duration nucleation phase
on slider 1 in the front of the $P$ wave on sider 2. After slider 2 stopped motion, slider
1 still moves and its peak velocity comes after that of slider 2. The occurrence time of
the peak velocity slightly increases with γ. Although a bump appears in the waveform
of slider 1, its peak velocity is much smaller than that of slider 2. Hence, unlike Fig. 5
there is almost only one event in the whole rupture process in Fig. 6. Meanwhile, the
maximum value of peak velocity of Fig. 6 is higher than that of Fig. 5. In the
right-hand-side panels of Fig. 6, the displacements of slider 1 (displayed by a solid
line) and slider 2 (displayed by a dashed line) appear almost simultaneously and
increase with time. The final displacement is higher on slider 2 than on slider 1, and
the difference between the two final displacements decreases with increasing γ.

Tal et al. (2018) who studied numerically the effects of fault roughness with

amplitude of $b_r$ on the nucleation process of earthquakes in the presence of a rat- and
state-dependent friction law. The roughness can yields local barriers and makes the





nucleation process complicated. They also found an increase in nucleation length with
$b_r$. Considering a broad weak zone with a locally strong asperity on a fault plane,
Shibazaki and Matsu'ura (1995) found that in the dynamic rupture of the asperity,
there are aseismic slip and foreshock or pre-event, depending on the peak stress of the
asperity, preceding the main rupture and the rupture of the asperity accelerates the
nucleation of main rupture. This study indicates the influence of heterogeneous
friction strengths on the generation of nucleation phase. Schmitt et al. (2015)
considered the importance of time-dependent stress heterogeneity on nucleation.
Although this factor is not taken into account in this study, the present study for
different values of $\phi$ on the two sliders seems able to meet the results obtained by the
three groups.

Secondly, we consider different values of $U_{c1}$ and $U_{c2}$. Simulation results are

displayed in Fig. 7 where the values of characteristic displacements are $U_{c1}$=0.5 and
$U_{c2}$=0.1 (with $\psi$=0.2) and the values of other model parameters are the same as those
in Fig. 6, i.e., $s$=0.48, $f_{o1}$=1.0 and $f_{o2}$=1.1 (with $\phi$=1.1), and (a) for $\gamma$=0.00 or $\eta_2$=0, (b)
for $\gamma$=0.01 or $\eta_2$=0.1, (c) for $\gamma$=0.05 or $\eta_2$=0.5, and (d) for $\gamma$=0.10 or $\eta_2$=1 when $\eta_1$=10.
Like Fig. 6, the left-hand-side panels show the existence of a very long-duration
nucleation phase on slider 1 in the front of the $P$ wave on sider 2. After slider 2
stopped motion, slider 1 still moves and its peak velocity comes after that of slider 2.
The occurrence time of the peak velocity slightly increases with $\gamma$. The maximum
value of peak velocity of Fig. 7 is higher than that of Fig. 6. In addition, the
predominant period of $P$ wave on slider 2 is shorter in Fig. 7 than in Fig. 6. This
might be due to a faster drop of friction force in Fig. 7 with shorter $U_{c2}$ than in Fig. 6
with longer $U_{c2}$. Although a peak velocity appears in the waveform of slider 1, its
amplitude is very much smaller than that of slider 2. Hence, unlike Fig. 5 there is



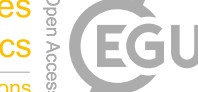
almost only one event in the whole rupture process of Fig. 7. In the right-hand-side
panels of Fig. 7, the displacement of slider 1 (displayed by a solid line) first appears
and increases with time; while the displacement of slider 2 (displayed by a dashed line)
suddenly appears for a while after slider 1 moves and then jumps to its peak value in a
short time. The final displacement is higher on slider 2 than on slider 1, and the
difference between the two final displacements decreases with increasing $\gamma$.

Using an infinite elastic model with a slip-dependent friction, Ionescu and

Campillo (1999) found the influence of the shape of the friction law and fault
finiteness on the duration of nucleation phase and the duration varies when the fault
length has the order of the characteristic length of the friction law. The present study
is essentially consistent with their results.

Thirdly, it is necessary to consider the effect on the simulations due to weak

seismic coupling (now $s$=0.17) between the two sliders when the values of other
model parameters are the same as those in Fig. 7, i.e., $f_{o1}$=1.0 and $f_{o2}$=1.1 (with $\phi$=1.1),
and $U_{c1}$=0.5 and $U_{c2}$=0.1 (with $\psi$=0.2) , and (a) for $\gamma$=0.00 or $\eta_2$=0, (b) for $\gamma$=0.01 or
$\eta_2$=0.1, (c) for $\gamma$=0.05 or $\eta_2$=0.5, and (d) for $\gamma$=0.10 or $\eta_2$=1 when $\eta_1$=10. Simulation
results are displayed in Fig. 8. Like Figs. 6 and 7, there is a very long-duration
nucleation phase on slider 1 in the front of the $P$ wave on sider 2. After slider 2
stopped motion, slider 1 still moves and its peak velocity comes after that on slider 2.
The peak velocity of slider 1 appears much later than that in Fig. 7. This might be due
to a fact that it needs a longer time to trigger slider 2 due to weak coupling between
the two sliders in Fig. 8. Meanwhile, the occurrence time of the peak velocity on
slider 2 slightly increases with $\gamma$. From the values of peak velocity as mentioned
above, the maximum value of peak velocity in Fig. 8 is higher than that in Fig. 7. This
indicates that weaker coupling between two sliders can yield a higher peak velocity

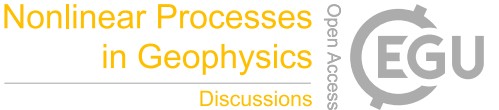



than stronger coupling. In addition, the predominant period of *P* wave on slider 2 is
shorter in Fig. 8 than in Fig. 7. Although a peak velocity appears in the waveform of
slider 1, its amplitude is very much smaller than that of slider 2. Unlike Fig. 5 there is
almost only one event in the whole rupture process of Fig. 8.
In the right-hand-side panels of Fig. 8, the displacement of slider 1 (displayed by
a solid line) first appears and increases with time; while the displacement of slider 2
(displayed by a dashed line) suddenly appears for a while after slider 1 moves and
then jumps to its peak value in a short time span. The final displacement is higher on
slider 2 than on slider 1, and the difference between the two final displacements
decreases with increasing $\gamma$.

4.4 Inertial Effect
The inertial effect (represented by $\mu$) on the earthquake nucleation is made for
different masses of the two sliders, i.e., $\mu>1$. Simulation results are displayed in Fig. 9
with $s=0.48$ and in Fig. 10 with $s=0.48$. In the two figures, the values of $\mu$ are: (a) for
$\mu=1$, (b) for $\mu=5$, (c) for $\mu=10$, and (d) for $\mu=30$ when $s=0.48$, $f_{o1}=1.0$ and $f_{o2}=1.1$
(with $\phi=1.1$), $U_{c1}=0.5$ and $U_{c2}=0.1$ (with $\psi=0.2$), and $\eta_1=10$ and $\eta_2=0$ (with $\gamma=0$).
Like Figs. 6−8, Figs. 9 and 10 show the existence of very long-duration nucleation
phases on slider 1 in the front of the *P* wave on sider 2. After slider 2 stopped moving,
slider 1 still moves and its peak velocity comes after that on slider 2. The occurrence
times of the peak velocity of both sliders 1 and 2 in Figs. 9 and 10 increase with $\mu$ and
are almost similar to those in Figs. 7 and 8, respectively. The occurrence times of the
peak velocity in Fig. 10 are longer than those in Fig. 9. This might be due to a fact
that a longer time is needed to trigger slider 2 due to weak coupling between the two
sliders in Fig. 10. Meanwhile, the predominant periods of the *P* wave on sider 2





increases with μ as expected. From the values of peak velocity as mentioned above,
the maximum value of peak velocity of Fig. 10 is higher than that of Fig. 9. This
indicates that that weaker coupling between two sliders can yield a higher peak
velocity on slider 2 than stronger coupling. In addition, the peak velocity on slider 1 is
lower than that on slider 2 and decreases with $\mu$ especially for small *s*. Although a
peak velocity appears in the waveform of slider 1 in Figs. 9 and 10, its amplitude is
much smaller than that of slider 1. Unlike Figure 5, there is almost only one event in
the rupture process in Figs. 9 and 10.

In Figs. 9 and 10, the peak velocity of *P* wave decreases with increasing $\mu$.

Numerical tests exhibit that the *P* wave almost becomes a nucleation phase on slider 2
when $\mu$>30. In the other word, the nucleation phase on slider 1 cannot trigger the *P*
wave on slider 2 when the mass of the latter is 30 times larger than that of the former.
When the densities and fault widths of the two sliders are equal, the fault length of
slider 2 is 30 times longer than that of slider 1 when $\mu$=30. Since the present model is
a strike-slip (SS) one, the empirical relationship of earthquake magnitude, *M*, versus
fault length, *L*, for the SS events is: *M=(5.16±0.13)+(1.12±0.08)log(L)* (Wells and
Coppersmith, 1994). When $\mu$=30 or $L_2$=30$L_1$, the related magnitudes are $M_1$ for slider
1 and $M_1$+1.65 for slider 2. This means that a nucleation phase with a magnitude of *M*
cannot trigger an earthquake with a magnitude of *M*+1.65.

In the right-hand-side panels of Figs. 9 and 10, the displacement of slider 1

(displayed by a solid line) first appears and increases with time; while the
displacement of slider 2 (displayed by a dashed line) suddenly appears for a while
after slider 1 moves and then jumps to its peak value in a short time span. The
difference in final displacement between the two sliders slightly increases with $\mu$ and

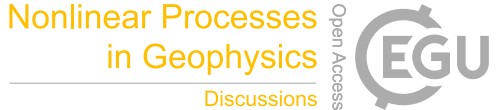

is bigger for small *s* than for large *s*. The phenomenon that the final displacement of
slider 1 is lower than that of slider 2 might be due to a fact that the force drop on
slider 2 is higher than that on slider 1.

4.5 Some Comparisons with Other Studies
Numerical simulations of this study exhibit that the ratios $\gamma=\eta_2/\eta_1$, $\phi=f_{o2}/f_{o1}$,
$\psi=U_{c2}/U_{c1}$, and $\mu=m_2/m_1$ are four important factors in influencing the earthquake
rupture processes including the generation of nucleation phase, yet the seismic
coupling s is a minor one. Except for the cases with equal values on the two sliders for
the four ratios, the nucleation phase happens on slider 1 and the *P* wave appears on
slider 2. When $\gamma>1$, $\phi=1$, $\psi=1$, and $\mu=1$, there is only a very short-duration nucleation
phase and the P wave appears very soon after the generation of nucleation phase. This
is inconsistent with Figure 1.

When $\gamma>1$, $\phi>1$, $\psi\geq1$, and $\mu=1$, there is a long-duration nucleation phase on

slider 1, the *P* wave appears on slider 2 much lately after the generation of nucleation
phase. Although the simulated waveforms are consistent with Fig. 1, the final
displacement of nucleation phase on slider 1 is the same as that of the *P* wave on
slider 2. This indicates equal values of total energy on the two sliders. It is
questionable, because the energy of nucleation phase is lower than that of the
mainshock from observations.

When $\gamma>1$, $\phi>1$, $\psi<1$, and $\mu=1$, the final displacement of nucleation phase is

smaller than that of *P* wave. The difference in the amplitudes between the *P* wave and
nucleation phase decreases with increasing *s*, increasing $\gamma$, or decreasing $\psi$. The
simulated waveforms are consistent with Fig. 1. The results are reasonable, because
the total energy on slider 1 is less than that on slider 2.





When $\gamma$>1, $\phi$>1, and $\psi$≪1, the peak velocity of slider 2 decreases with increasing
$\mu$, and becomes very small when $\mu$>30, even though the final displacement of
nucleation phase is still smaller than that of *P* wave. The degree of similarity of
simulated waveforms of these cases (see Figs. 9 and 10) with Fig. 1 decreases with
increasing $\mu$. The upper-bound value of $\mu$ to yield transition from nucleation phase to
the *P* wave from observations is 30. Consequently, the optimal conditions for
generating the nucleation phase on slider 1 plus the *P* wave on slider 1 as displayed in
Figure 1 and the results from other studies are $\gamma$>1, $\phi$>1, $\psi$<1, and $\mu$<30. Of course,
there are upper-bound values for $\gamma$ and $\phi$ and a lower-bound value for $\psi$ as mentioned
in the last section. Note that the upper-bound value of a ratio depends on the values of
other ratios.
However, a difference between the present study and previous ones is that the
nucleation phase appears on slider 1 and does not disappear after the presence of *P*
wave on slier 2. This might be due to a use of a two-body model in this study and uses
of a one-body or 1-D model is taken in others. Meanwhile, the mechanism (including
friction and viscosity) to yield the transition from quasi-static motions to dynamic
ruptures proposed in this study is the same as that in Wang (2017a), yet different from
others who only considered the frictional effect. However, unlike Wang (2017a) the
present simulation results cannot lead to the conclusion that the peak amplitude of *P*
wave, which is associated with the earthquake magnitude, is independent upon the
duration time of nucleation phase. In addition, the inertial effect was not taken into
account by Wang (2017a).
Based on an infinite elastic model with slip-dependent friction, Shibazaki and
Matsu'ura (1992) assumed that the transition process includes three phases: phase-I
for the low quasi-static nucleation, phase-II for the onset of dynamic ruptured with

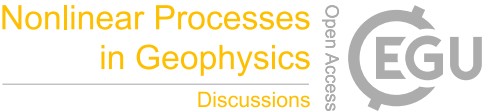

slow rupture growth in the absence of seismic-wave radiation, and phase-III for the
high-speed rupture propagation with seismic-wave radiation. Shibazaki and Matsu'ura
(1993) further found that the accelerating stage from phase-II to phase-III is related to
the presence of nucleation phase in the front of the main $P$ wave. Their results are
similar to those obtained by Ueda et al. (2014, 2015) and Kawamura et al (2018). The
results of this study and Wang (2017a) only show two stages which are comparable
with the phase-II and phase-II stages proposed by Shibazaki and Matsu'ura (1992,
1993). From the analytic solutions of an infinite elastic model with a slip-dependent
friction, Campillo and Ionescu (1997) expressed how the initiation phase determines
the transition to the P wave and claimed that the transition is controlled by an
apparent supersonic velocity of the rupture front. However, the present result does not
seem to meet their conclusion. According to an infinite elastic model with rate- and
state-dependent friction, Segall and Rice (2006) divided the weakening processes of
ruptures into the nucleation regime dominated by rate and state frictional weakening
and a transition regime to thermal pressurization. In the present study, the thermal-
pressurized slip-weakening friction is considered during the whole rupture process
and the results show a transition from the nucleation phase with smaller $f_{o1}$ and $U_{c1}$
on slider 1 to the $P$ wave with larger $f_{o2}$ and $U_{c2}$ on slider 2. Hence, the present result
could be only partly consistent with their conclusion.

**5 Conclusions**
We study the frictional and viscous effects on earthquake nucleation based on a
two-body spring-slider model in the presence of thermal-pressurized slip-dependent
friction and viscosity. The stiffness ratio of the system is the ratio of coil spring $K$
between two sliders and the leaf spring $L$ between a slider and the background plate




and denoted by $s=K/L$. The $s$ is not a significant factor in generating the nucleation
phase. The masses of the two sliders are $m_1$ and $m_2$, respectively. The frictional and
viscous effects are specified by the static friction force, $f_o$, the characteristic
displacement, $U_c$, and viscosity coefficient, $\eta$, respectively. Simulation results show
that friction and viscosity can both lengthen the natural period of the system and
viscosity increases the duration time of motion of the slider. Higher viscosity causes
lower particle velocities than lower viscosity. The ratios $\gamma=\eta_2/\eta_1$, $\phi=f_{o2}/f_{o1}$,
$\psi=U_{c2}/U_{c1}$, and $\mu=m_2/m_1$ are four important factors in influencing the generation of a
nucleation phase. When $\gamma>1$, $\phi=1$, $\psi=1$, and $\mu=1$, the nucleation phase is generated on
slider 1 and the $P$ wave appear on slider 2. But, the $P$ wave appears very soon after
the generation of nucleation phase. When $\gamma>1$, $\phi>1$, $\psi\geq1$, and $\mu=1$, the $P$ wave
appears much lately after the generation of nucleation phase. When $\psi\geq1$, the final
displacement of nucleation phase is almost equal to that of $P$ wave. When $\psi<1$, the
final displacement of nucleation phase is smaller than that of $P$ wave. The difference
in the amplitudes between the $P$ wave and nucleation phase decreases when either $s$ or
$\gamma$ increases and $\psi$ decreases. The peak velocity of $P$ wave on slider 2 decays with
increasing $\mu$, thus suggesting that the inertial effect is important on the rupture
processes. Consequently, when $s>0.17$, $\gamma>1$, $1.15>\phi>1$, $\psi<1$, and $\mu<30$ simulation
results exhibit the generation of nucleation phase on slider 1 and the formation of $P$
wave on slider 2. The results are consistent with the observations and suggest the
possibility of generation of nucleation phase on a sub-fault. This answer the question
pointed out in this study.

*Acknowledgments*. The study was by Academia Sinica, the Ministry of Science and
Technology, and the Central Weather Bureau, TAIWAN for financial support.

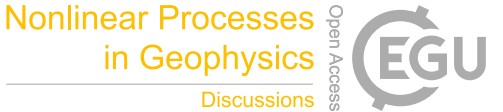

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






**List of Figure Captions**
Figure 1. An example to show the nucleation phase, onset of the $P$ wave, and the $P$
wave in velocity seismogram.
Figure 2. A two-body spring-slider model: $F_i$=the friction force at the $i$-th slider,
$m_i$=the mass of the $i$-th slider, $K$=the stiffness between two sliders, $L_i$=the
stiffness between the $i$-th slider and the moving plate, $C_i$=the viscosity
coefficient between the $i$-th slider and the moving plate, and $v_p$=the velocity of
the moving plate, and $u_i$ ($i$=1, 2) is the displacement of the $i$-th slider.
Figure 3. The variations of friction force with sliding displacement for $u_c$=0.1, 0.3, 0.5,
0.7, and 0.9 m when $F_o$=1 unit.
Figure 4. The time sequences of $V/V_{max}$ and $U/U_{max}$: (a) for $s$=0.06, (b) for $s$=0.12, (c)
for $s$=0.30, and (d) for $s$=0.48 when $\mu$=1, $f_{o1}$=1.0 and $f_{o2}$=1.0 (with $\phi$=1), $U_{c1}$=0.5
and $U_{c2}$=0.5 (with $\psi$=1), and $\eta_1$=0 and $\eta_2$=0 (with $\gamma$=1).
Figure 5. The time sequences of $V/V_{max}$ and $U/U_{max}$: (a) for $\gamma$=0.00, (b) for $\gamma$=0.01, (c)
for $\gamma$=0.05, and (d) for $\gamma$=0.10 when $s$=0.48, $\mu$=1, $\eta_1$=10, $f_{o1}$=1.0 and $f_{o2}$=1.0
(with $\phi$=1), and $U_{c1}$=0.5 and $U_{c2}$=0.5 (with $\psi$=1).
Figure 6. The time sequences of $V/V_{max}$ and $U/U_{max}$: (a) for $\gamma$=0.00, (b) for $\gamma$=0.01, (c)
for $\gamma$=0.05, and (d) for $\gamma$=0.10 when $s$=0.48, $\mu$=1, $\eta_1$=10, $f_{o1}$=1.0 and $f_{o2}$=1.1
(with $\phi$=1.1), and $U_{c1}$=0.5 and $U_{c2}$=0.5 (with $\psi$=1).
Figure 7. The time sequences of $V/V_{max}$ and $U/U_{max}$: (a) for $\gamma$=0.00, (b) for $\gamma$=0.01, (c)
for $\gamma$=0.05, and (d) for $\gamma$=0.10 when $s$=0.48, $\mu$=1, $\eta_1$=10, $f_{o1}$=1.0 and $f_{o2}$=1.1
(with $\phi$=1.1), and $U_{c1}$=0.5 and $U_{c2}$=0.1 (with $\psi$=0.2).
Figure 8. The time sequences of $V/V_{max}$ and $U/U_{max}$: (a) for $\gamma$=0.00, (b) for $\gamma$=0.01, (c)
for $\gamma$=0.05, and (d) for $\gamma$=0.10 when $s$=0.17, $\mu$=1, $\eta_1$=10, $f_{o1}$=1.0 and $f_{o2}$=1.1





(with $\phi$=1.1), and $U_{c1}$=0.5 and $U_{c2}$=0.1 (with $\psi$=0.2).

Figure 9. The time sequences of $V/V_{max}$ and $U/U_{max}$: (a) for $\mu$=1, (b) for $\mu$=5, (c) for

$\mu$=10, and (d) for $\mu$=30 when $s$=0.48, $f_{o1}$=1.0 and $f_{o2}$=1.1 (with $\phi$=1.1), $U_{c1}$=0.5

and $U_{c2}$=0.1 (with $\psi$=0.2), and $\eta_1$=10 and $\eta_2$=0 (with $\gamma$=0).

Figure 10. The time sequences of $V/V_{max}$ and $U/U_{max}$: (a) for $\mu$=1, (b) for $\mu$=5, (c) for

$\mu$=10, and (d) for $\mu$=30 when $s$=0.17, $f_{o1}$=1.0 and $f_{o2}$=1.1 (with $\phi$=1.1), $U_{c1}$=0.5

and $U_{c2}$=0.1 (with $\psi$=0.2), and $\eta_1$=10 and $\eta_2$=0 (with $\gamma$=0).








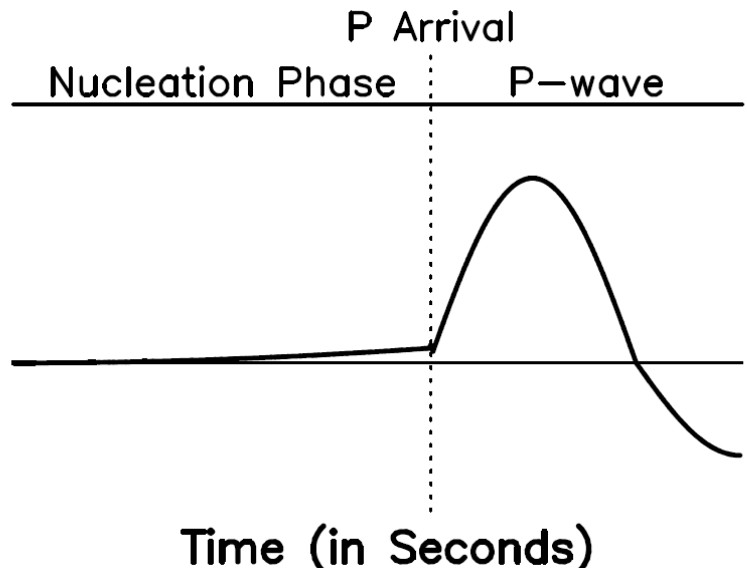


Figure 1. An example to show the nucleation phase, onset of the *P* wave, and the *P*
wave in velocity seismogram.






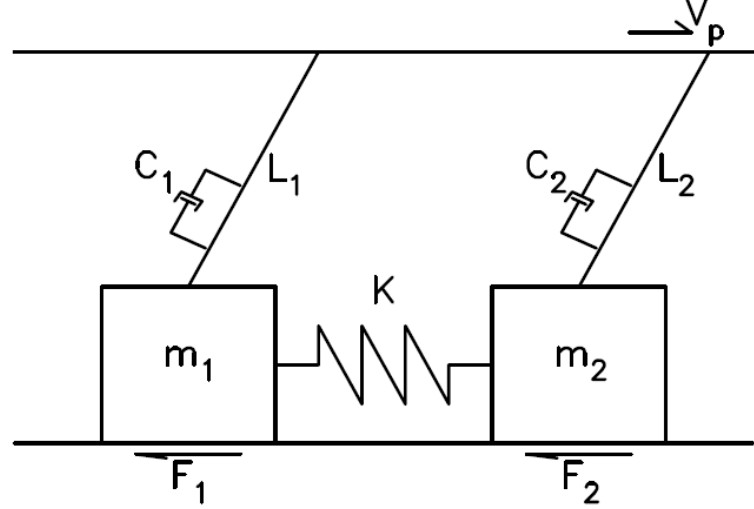


**Figure 2**. A two-body spring-slider model: $F_i$=the friction force at the $i$-th slider, $m_i$=the mass of the $i$-th slider, $K$=the stiffness between two sliders, $L_i$=the stiffness between the $i$-th slider and the moving plate, $C_i$=the viscosity coefficient between the $i$-th slider and the moving plate, and $v_p$=the velocity of the moving plate, and $u_i$ ($i$=1, 2) is the displacement of the $i$-th slider.









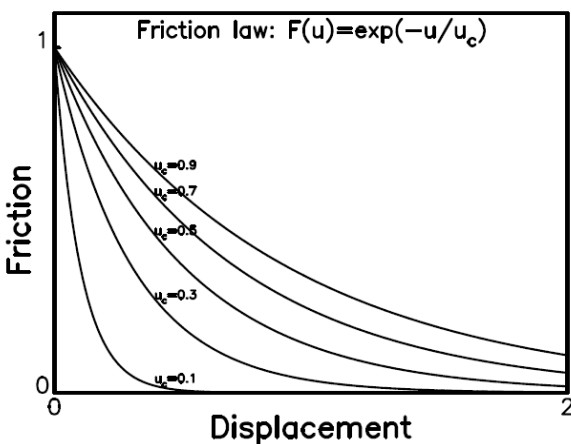


**Figure 3**. The variations of friction force with sliding displacement for $u_c$=0.1, 0.3,
0.5, 0.7, and 0.9 m when $F_o$=1 unit.






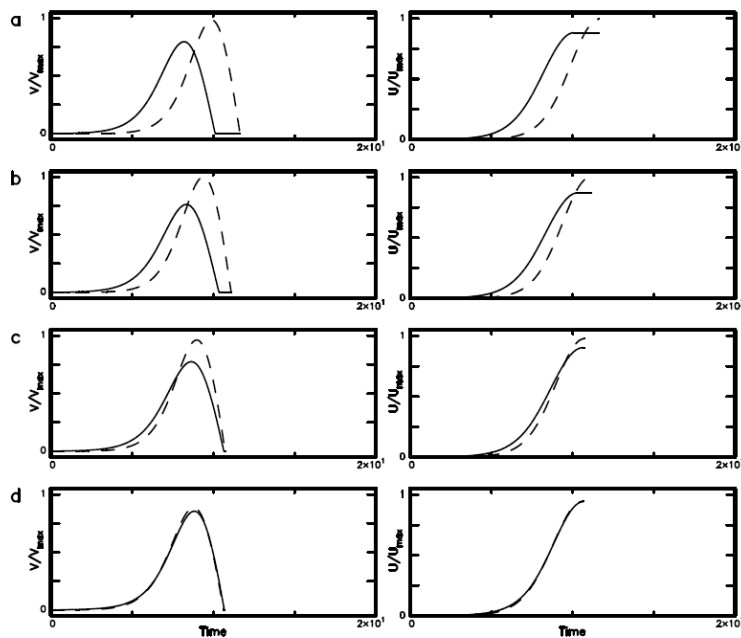


**Figure 4**. The time sequences of $V/V_{max}$ and $U/U_{max}$: (a) for $s$=0.06, (b) for $s$=0.12, (c)
for $s$=0.30, and (d) for $s$=0.48 when $\mu$=1, $f_{o1}$=1.0 and $f_{o2}$=1.0 (with $\phi$=1), $U_{c1}$=0.5 and
$U_{c2}$=0.5 (with $\psi$=1), and $\eta_1$=0 and $\eta_2$=0 (with $\gamma$=1).










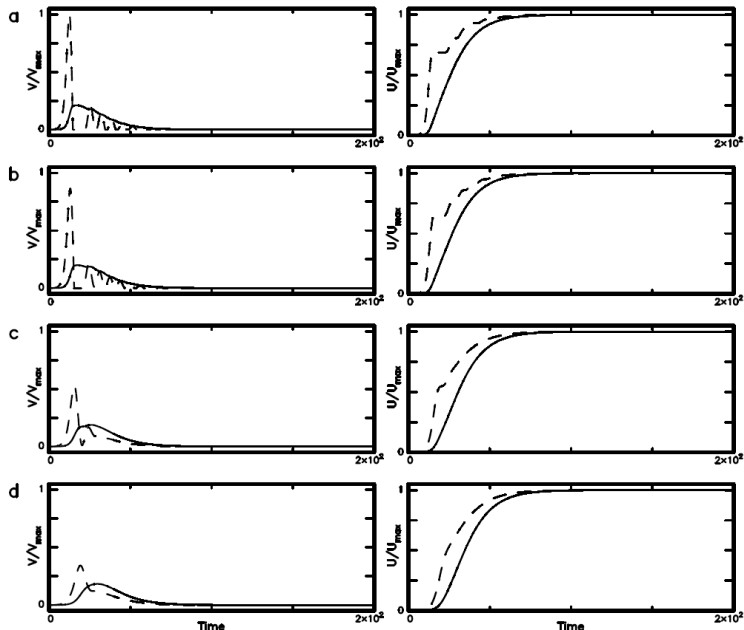


**Figure 5**. The time sequences of $V/V_{max}$ and $U/U_{max}$: (a) for $\gamma$=0.00, (b) for $\gamma$=0.01, (c) for $\gamma$=0.05, and (d) for $\gamma$=0.10 when $s$=0.48, $\mu$=1, $\eta_1$=10, $f_{o1}$=1.0 and $f_{o2}$=1.0 (with $\phi$=1), and $U_{c1}$=0.5 and $U_{c2}$=0.5 (with $\psi$=1).







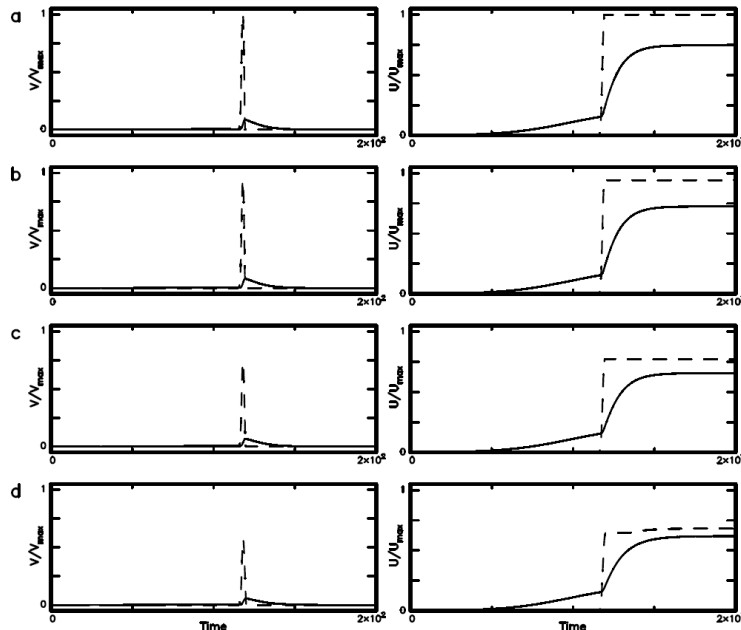


**Figure 6**. The time sequences of $V/V_{max}$ and $U/U_{max}$: (a) for $\gamma$=0.00, (b) for $\gamma$=0.01, (c) for $\gamma$=0.05, and (d) for $\gamma$=0.10 when s=0.48, $\mu$=1, $\eta_1$=10, $f_{o1}$=1.0 and $f_{o2}$=1.1 (with $\phi$=1.1), and $U_{c1}$=0.5 and $U_{c2}$=0.5 (with $\psi$=1).










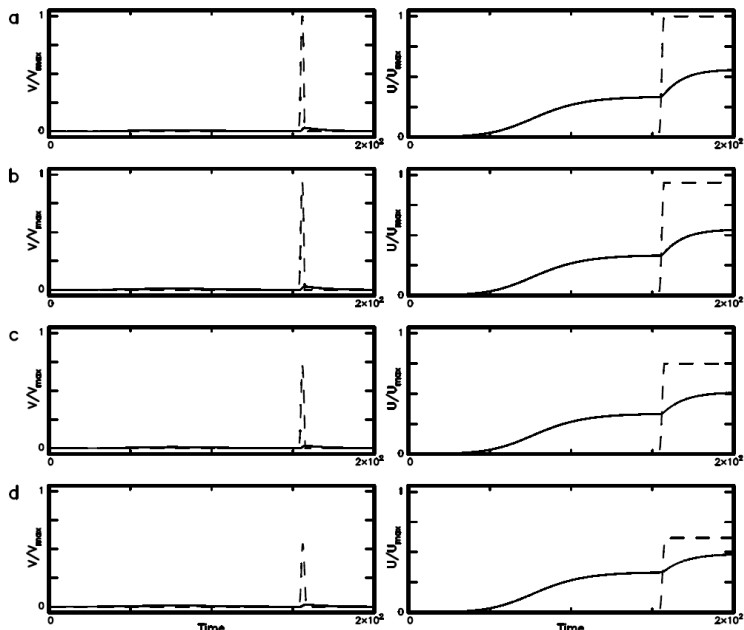


**Figure 7**. The time sequences of $V/V_{max}$ and $U/U_{max}$: (a) for $\gamma$=0.00, (b) for $\gamma$=0.01, (c)
for $\gamma$=0.05, and (d) for $\gamma$=0.10 when s=0.48, $\mu$=1, $\eta_1$=10, $f_{o1}$=1.0 and $f_{o2}$=1.1 (with
$\phi$=1.1), and $U_{c1}$=0.5 and $U_{c2}$=0.1 (with $\psi$=0.2).







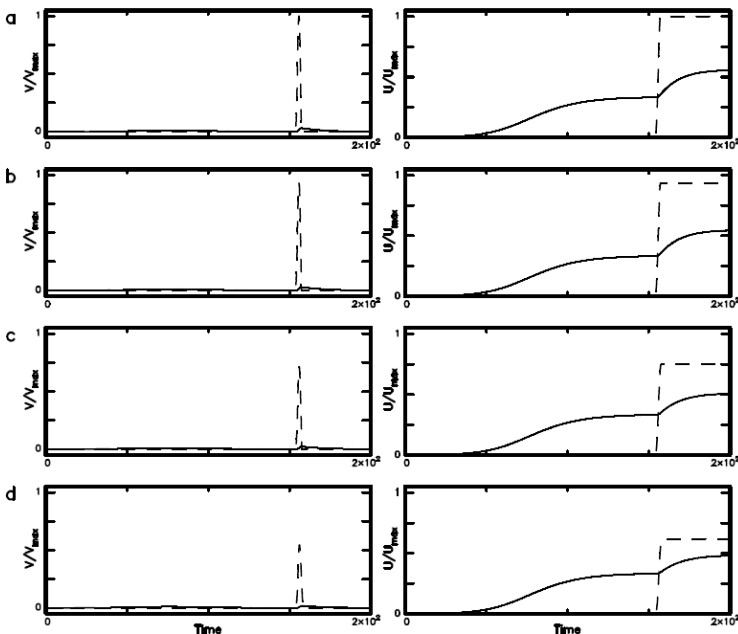


**Figure 8**. The time sequences of $V/V_{max}$ and $U/U_{max}$: (a) for $\gamma$=0.00, (b) for $\gamma$=0.01, (c) for $\gamma$=0.05, and (d) for $\gamma$=0.10 when $s$=0.17, $\mu$=1, $\eta_1$=10, $f_{o1}$=1.0 and $f_{o2}$=1.1 (with $\phi$=1.1), and $U_{c1}$=0.5 and $U_{c2}$=0.1 (with $\psi$=0.2).










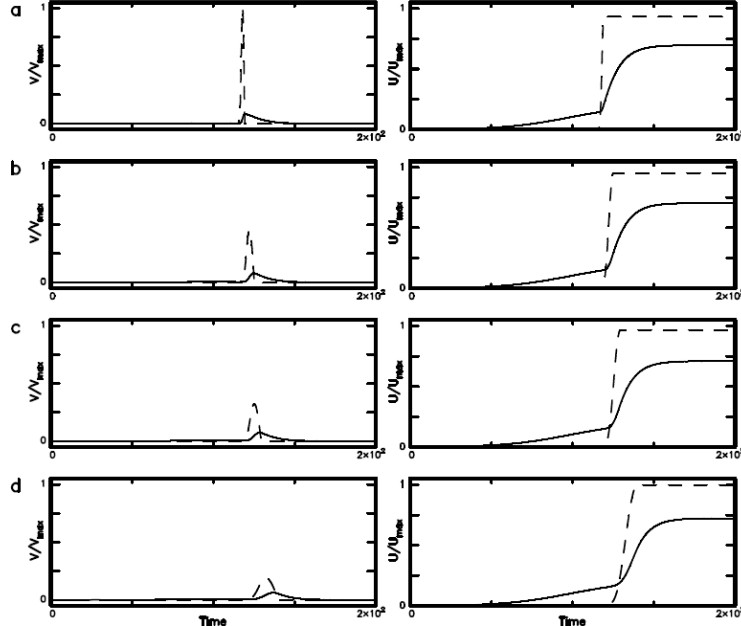


**Figure 9**. The time sequences of $V/V_{max}$ and $U/U_{max}$: (a) for $\mu=1$, (b) for $\mu=5$, (c) for
$\mu=10$, and (d) for $\mu=30$ when $s=0.48$, $f_{o1}=1.0$ and $f_{o2}=1.1$ (with $\phi=1.1$), $U_{c1}=0.5$ and
$U_{c2}=0.1$ (with $\psi=0.2$), and $\eta_1=10$ and $\eta_2=0$ (with $\gamma=0$).







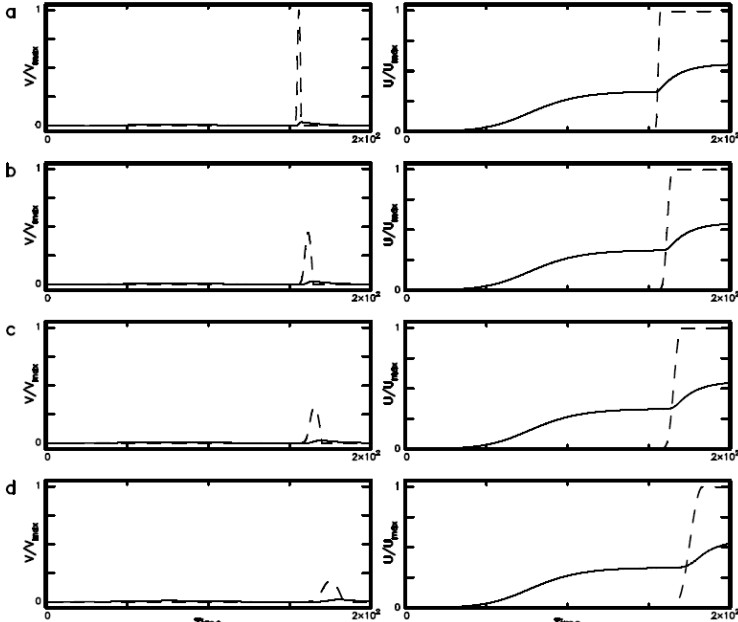


**Figure 10**. The time sequences of $V/V_{max}$ and $U/U_{max}$: (a) for $\mu$=1, (b) for $\mu$=5, (c) for $\mu$=10, and (d) for $\mu$=30 when $s$=0.17, $f_{o1}$=1.0 and $f_{o2}$=1.1 (with $\phi$=1.1), $U_{c1}$=0.5 and $U_{c2}$=0.1 (with $\psi$=0.2), and $\eta_1$=10 and $\eta_2$=0 (with $\gamma$=0).
