# Peer review of "Can the Nucleation Phase be Generated on a Sub-fault"

_Nonlinear Processes in Geophysics, 2018_

## Referee Comment (RC1) · Anonymous Referee #1 · 28 Feb 2019

The presented work is devoted to the vital question of active fault dynamics. Discussed are the process of earthquake nucleation in the presence of thermal-pressurized slip-dependent friction and viscosity. The main question is pointed in introduction: Can the nucleation phase happen on a sub-fault which links to the main fault of an earthquake? The author has performed numerical experiments on the basis of two-body spring-slider model. The paper studies the influence of a large number of parameters on the dynamics of system of 2 blocks and the peculiarities of their slips. The author shows that under certain conditions there is a sequence of slow movement of one block at the beginning, followed by a rapid slip of the second block. The assumption is put forward in the work that this effect can account for the generation of nucleation phase on a sub-fault. General issues: I think that using such simple models has to be accurately

grounded, and even more arguments are needed to apply the obtained results to real processes taking place in natural fault zones. In a system consisting of two blocks, almost always the slippage of one block will trigger a fast (or slow) slip of the other. Currently, there is a large number of works on the dynamics of multi-block slider-model, including a large number of works in NPG, for example: https://doi.org/10.5194/npg-24-215-2017, and presented in the introduction. They tend to present a very complex system dynamics. The article of J.-H. Wang does not have any benefits and has a number of serious simplifications. 2. There are many "descriptions" in this article. Almost all come down to a description of how the block moves, it does not give any time variations of velocity and another relationship/ Moreover, the pictures are made in very poor quality, which makes it difficult to understand the features of the process. It worth mentioning, that the main assumption is presented in Figure 1, where first is a phase of linear growth followed by a dynamic slip. In any cases presented in this article this characteristic behavior is not observed. 3. One gets the impression that due to very serious simplifications of the numerical model, the discussion is reduced to a detailed description of all possible realization and occasionally a comparison with other works is given. But, presented results coinciding only partially with field observations and the numerical experiments. In addition, it was worth adding a discussion on the influence of slow slip events on the generation of large earthquakes, which in my opinion is more applicable to this work.

In view of all the remarks above, I think that the paper doesn't contain any new and significant results and can't be published in Nonlinear Process in Geophysics.

Please also note the supplement to this comment:
https://www.nonlin-processes-geophys-discuss.net/npg-2018-49/npg-2018-49-RC1-supplement.pdf

———————————————————

---

## Author Comment (AC1) · 4 Mar 2019

The author has performed numerical experiments on the basis of two-body spring slider model. The paper studies the influence of a large number of parameters on the dynamics of system of 2 blocks and the peculiarities of their slips. The author shows that under certain conditions there is a sequence of slow movement of one block at the beginning, followed by a rapid slip of the second block. The assumption is put forward in the work that this effect can account for the generation of nucleation phase on a sub-fault. General issues: I think that using such simple models has to be accurately grounded, and even more arguments are needed to apply the obtained results to real processes taking place in natural fault zones. In a system consisting of two blocks, almost always the slippage of one block will trigger a fast (or slow) slip of the other. Currently, there is

a large number of works on the dynamics of multi-block slider-model, including a large number of works in NPG, for example: https://doi.org/10.5194/npg-24-215-2017, and presented in the introduction. They tend to present a very complex system dynamics. The article of J.-H. Wang does not have any benefits and has a number of serious simplifications. [Answer] Since 1967 when Burridge and Knopoff proposed their multi-block spring slider model, there have been a large number of works on the dynamics of the model. I myself have also studied numerous seismological problems based on the model. The studies about generation of nucleation phase and initiation of dynamic slip (or an earthquake) on a single fault can be seen in Wang (J. Seismol. 2017). Since the present study concentrates on the dynamics of generation of nucleation phase on a sub-fault and initiation of dynamic slip (or an earthquake) on a main fault, a two-body spring-slider model is taken into account. Of course, a multi-block spring slider model can provide more information. Nevertheless, it is easier clearer to explore the interaction between nucleation phase on a sub-fault and main dynamic slip on the main fault by using a two-block spring-slider model.

There are many "descriptions" in this article. Almost all come down to a description of how the block moves, it does not give any time variations of velocity and another relationship. Moreover, the pictures are made in very poor quality, which makes it difficult to understand the features of the process. It worth mentioning, that the main assumption is presented in Figure 1, where first is a phase of linear growth followed by a dynamic slip. In any cases presented in this article this characteristic behavior is not observed. [Answer] The "descriptions" given in the text are just written to explain simulation results, with a focus on the interaction between the nucleation phase on the sub-fault and dynamic slip (or an earthquake) on the main fault. Hence, the description about the time variations in velocities and displacements is not the major one of the study. Of course, I can add some statements to describe the time variations in velocities and displacements and their relationships in the revised manuscript after the Editor allow me to submit the revised version. In Figures 4 and 5, there are not clear nucleation phases on the sub-fault. In Figures $6-10$, we can see the nucleation phase

which grows linearly with time on the sub-fault and is followed by dynamic slip on the main fault. This is essentially consistent with Figure 1.

One gets the impression that due to very serious simplifications of the numerical model, the discussion is reduced to a detailed description of all possible realization and occasionally a comparison with other works is given. But, presented results coinciding only partially with field observations and the numerical experiments. In addition, it was worth adding a discussion on the influence of slow slip events on the generation of large earthquakes, which in my opinion is more applicable to this work. [Answer] I think simulation results of this study can help us to understand two things: (1) the nucleation phase being able to be generated on a sub-fault linked to the main fault of an earthquake; and (2) the major physical factors in controlling the processes. Of course, it is OK for me to add more statements to describe the influence of slow slip on the initiation of large earthquake in the revised manuscript after the Editor allow me to submit the revised version.

Please also note the supplement to this comment:
https://www.nonlin-processes-geophys-discuss.net/npg-2018-49/npg-2018-49-AC1-supplement.pdf

---

## Referee Comment (RC2) · Anonymous Referee #2 · 6 Mar 2019

The manuscript investigates the influence of different phenomena thought to be related to the frictional fault sliding on the emergence of the nucleation phase. To this end dynamics of a two block sliding model system with viscosity and displacement-dependent dry friction is analysed. The first thing which is apparent from the model equations is that frictional sliding is assumed to be always present, that is the system is always in the limiting stage, which is a great oversimplification. The second feature is that the friction is assumed to depend upon displacement rather than velocity as conventionally accepted. Furthermore, the model has a lot of parameters, so it is not surprising that some combination of parameters does produce the behaviour resembling the nucleation phase. The question is then as to why nature is reduced to these combinations of parameters.

The numerical solution was not verified against the particular cases which either allow analytical solutions or could be referred to existing numerical solutions, so there is no way to believe in the correctness of the model.

For these reasons the proposed model does not seem to have any value.

The manuscript is badly written, English is substandard and the terminology is sometimes confused. For instance spring stiffness (elastic parameter) is sometimes called stiffness strength (failure parameter).

I cannot recommend the manuscript for publication.

---

## Author Comment (AC2) · 7 Mar 2019

Response to Comments by Reviewer #2

Although you cannot accept my manuscript, I still want to say thanks to you for valuable comments.

The manuscript investigates the influence of different phenomena thought to be related to the frictional fault sliding on the emergence of the nucleation phase. To this end dynamics of a two block sliding model system with viscosity and displacement-dependent dry friction is analysed. The first thing which is apparent from the model equations is that frictional sliding is assumed to be always present, that is the system is always in the limiting stage, which is a great oversimplification. [Answer] For most of studies on

dynamics of earthquake ruptures, frictional sliding is assumed to be always present. This is not my own selection.

The second feature is that the friction is assumed to depend upon displacement rather than velocity as conventionally accepted. Furthermore, the model has a lot of parameters, so it is not surprising that some combination of parameters does produce the behavior resembling the nucleation phase. The question is then as to why nature is reduced to these combinations of parameters. [Answer] In my past studies, I very often applied velocity-dependent friction to dynamical modelling of earthquake ruptures based on spring-slider models. There are two reasons why slip-dependent friction is used in this study: (1) Madariaga and Cochard (1994) pointed out that purely velocity-dependent friction could yield unphysical phenomena and mathematically ill-posed problems and Ohnaka (2003) also stressed that the pure rate-dependent friction law is not a one-valued function of velocity. (2) The slip-dependent friction law comes from the end-member model, i.e., the adiabatic, undrained deformation (AUD) model, due to thermal pressurization model in the fault zone proposed by Rice (2006). (The other end-member model is the slip on a plane (SOP) model. In the AUD model, the sliding slip is dependent upon the sliding velocity which is not constant; while in the SOP model, the sliding velocity is always constant during the rupture processes. In the present study, I used the AUD model, and thus slip-dependent friction is somewhat velocity-dependent. References: Madariaga, R. and A. Cochard (1994). Seismic source dynamics, heterogeneity and friction. Ann. Geofis., 37(6), 1349-1375) Ohnaka, M. (2003). A constitutive scaling law and a unified comprehension for frictional slip failure, shear fracture of intact rocks, and earthquake rupture. J. Geophys. Res., 108, B2, 2080, doi:10.1029/ 2000JB000123.

The numerical solution was not verified against the particular cases which either allow analytical solutions or could be referred to existing numerical solutions, so there is no way to believe in the correctness of the model. [Answer] The analytic solutions were made only for discussing the predominant periods of two sliders. Numerical solutions

are essentially consistent with the analytic ones.

Please also note the supplement to this comment:
https://www.nonlin-processes-geophys-discuss.net/npg-2018-49/npg-2018-49-AC2-supplement.pdf